# ENHANCING 3D HUMAN POSE ESTIMATION WITH A LEARNED STRUCTURE-PRESERVING LOSS

## ABSTRACT

3D human pose estimation (3D HPE) is a challenging task due to complex structural constraints that are not well captured by standard training objectives such as mean squared error (MSE). Previous studies have attempted to enforce structural consistency by incorporating manually designed priors, rule-based constraints, or specialized architectures, which often limit adaptability. In this paper, we propose SCoTL-pose (Structural Consistency via Trainable Loss for Pose Estimation) framework that enables pose estimation models (pose-net) to learn structural dependencies directly from data, through a trainable loss function (loss-net), without explicit priors. Our approach introduces a graph-based loss-net that captures both local and global joint relationships, ensuring anatomically plausible pose predictions. While inspired by the idea of Structured Energy As Loss (SEAL), we extend it to tackle 3D human pose estimation, a task with more complex and high-dimensional structural dependencies than those considered in previous applications. To this end, we employ a graph-based model as loss-net architecture, tailored to capturing the intricate local and global dependencies among joints. SCoTL-pose can be combined with diverse backbones, from single-frame lifting networks to state-of-the-art multi-frame temporal models, without additional inference cost. To assess whether SCoTL-pose enhances structural plausibility in a quantitative manner, we also introduce Limb Symmetry Error (LSE) and Body Segment Length Error (BSLE) as evaluation metrics. Experimental results on Human3.6M, MPI-INF-3DHP, and Human3.6M WholeBody datasets demonstrate that SCoTL-pose not only reduces per-joint pose estimation errors but also generates more plausible poses, with increasing gains under more challenging settings such as single-frame or in-the-wild scenarios.

## 1 INTRODUCTION

3D human pose estimation (3D HPE) requires predicting accurate joint positions while preserving the underlying anatomical structure (Liu et al., 2024). This task is particularly difficult because the output space is governed by complex local and global dependencies between joints. However, common training objectives such as mean squared error (MSE) and mean per-joint position error (MPJPE) penalize individual joint errors without accounting for structural consistency, which often results in implausible or anatomically inconsistent poses. Therefore, it is critical to effectively model the structures in the output space to predict accurate and plausible 3D poses. Previous studies (Zheng et al., 2020; Wu et al., 2022; Fang et al., 2018; Xu et al., 2022; Kim & Lee, 2024) have attempted to capture such structural dependencies, but they are often constrained by manually designed rules or architecture-specific designs, which limit scalability and adaptability across different models.

To overcome these limitations, we propose SCoTL-pose (Structural Consistency via Trainable Loss for Pose Estimation), a novel framework that employs a trainable loss function to provide structural guidance for 3D pose estimation without requiring explicit priors. At the core of this framework is the loss-net, a neural network jointly optimized with the pose estimation model (pose-net). The loss-net learns to capture dependencies among joints and dynamically evaluates pose plausibility during training, unlike conventional per-joint error objectives. Building on the Structured Energy As Loss (SEAL) framework (Lee et al., 2022), initially applied to generic multi-label classification problems and natural language applications, we extend the concept of trainable loss functions to the 3D HPE problem, which involves more complex and high-dimensional structural dependencies.

To tackle structural challenges of 3D HPE, we design the loss-net as a graph-based model to better capture output structures of human poses, well-suited for capturing the intricate local and global structural dependencies. Moreover, to evaluate the structure of predicted poses, we introduce two complementary evaluation metrics, Limb Symmetry Error (LSE) and Body Segment Length Error (BSLE), which measure the structural consistency beyond traditional error metrics such as MPJPE. Our framework is model-agnostic and can be adaptable to various scenarios such as single-frame and whole-body pose estimation while supporting recent methods in multi-frame settings, because it only requires adding the loss-net in the training procedure. Moreover, SCoTL-pose does not introduce additional inference cost at test time since the loss-net is only utilized during training.

Extensive experiments on Human3.6M (Ionescu et al., 2014), MPI-INF-3DHP (Mehta et al., 2017), and Human3.6M 3D WholeBody (Zhu et al., 2023) demonstrate that SCoTL-pose not only reduces per-joint errors (Table 1,2, 3) but also produces more plausible poses, evaluated by the proposed LSE and BSLE (Table 4 and Figure 2, 3). In particular, it shows greater benefits in more challenging settings such as single-frame and in-the-wild dataset, where explicit priors are insufficient. Our results underscore the value of trainable loss functions in modeling complex structural dependencies and suggest a promising direction for a wide range of applications in structured prediction tasks.

## 2 RELATED WORK

### 2.1 3D HUMAN POSE ESTIMATION

3D human pose estimation is a well-established computer vision task involving the prediction of 3D joint positions from 2D images or videos. This task is inherently challenging because it requires inferring spatial relationships while ensuring anatomical plausibility from incomplete visual information. Current approaches generally follow two paradigms: (1) directly predicting 3D poses from images (Pavlakos et al., 2017; 2018) or (2) estimating 2D poses first and then lifting them to 3D space (Zheng et al., 2023; Liu et al., 2024). The 2D-to-3D lifting has been widely adopted due to the progress in 2D human pose estimation (Zheng et al., 2023).

More recently, there has been increasing emphasis on utilizing temporal information for 3D HPE. Multi-frame models, such as P-STMO (Shan et al., 2022), MixSTE (Zhang et al., 2022), Pose-Former (Zheng et al., 2021; Zhao et al., 2023), leverage sequences of frames and achieve stronger performance by exploiting richer spatial-temporal information. On the other hand, traditional single-frame settings remain relatively ambiguous due to the absence of temporal context, making them more challenging but also valuable for evaluating the robustness of new approaches.

Another recent direction is 3D whole-body pose estimation. For example, the Human3.6M 3D WholeBody (H3WB) dataset (Zhu et al., 2023) extends the widely used Human3.6M dataset by providing annotations for 133 keypoints, including those for the face, hands, and feet. Whole-body datasets have become an important benchmark, encouraging methods that move beyond traditional body keypoints toward more fine-grained and comprehensive human pose estimation.

### 2.2 OUTPUT STRUCTURE OF 3D HPE

3D HPE has inherent challenges such as ambiguity due to incomplete information, which is further compounded in single-frame scenarios. To address this issue, previous works have designed methods that consider prior knowledge of human body structure and joint relationships (Fang et al., 2018; Xu et al., 2022; Zheng et al., 2020; Chen et al., 2022), and enforced structural plausibility by explicitly constraining bone length, angles, and symmetry (Cao & Zhao, 2021; Wu et al., 2022; Bigalke et al., 2022; Chen et al., 2022).

Beyond these approaches, recent studies have explored generating multiple hypotheses or plausible 3D poses to alleviate depth ambiguity and structural uncertainty. For instance, Kim & Lee (2024) proposed a Biomechanical Pose Generator to augment training data with biomechanically valid poses, along with Binary Depth Coordinates to resolve the depth ambiguity by classifying the joint depths as front or back. Similarly, Rommel et al. (2024) introduced ManiPose, a manifold-constrained multi-hypothesis approach that estimates the plausibility of each candidate and restricts them to the human pose manifold.

Despite their contributions, most existing methods rely on prior knowledge or predefined rules, which may limit scalability and adaptability. In contrast, we aim to address these limitations by providing a more flexible and general approach for 3D HPE that captures joint dependencies without explicit prior knowledge. In addition, our method, SCoTL-pose, is agnostic to model architecture and can potentially be extended to various tasks with complex output structures.

### 2.3 TRAINABLE LOSS FUNCTION

Our idea of a trainable loss function builds on the Structured Energy As Loss (SEAL) (Lee et al., 2022). SEAL introduced a structured energy network as a trainable loss, showing that it can model complex dependencies and provide better supervision than static objectives. However, its prior applications were limited to probabilistic models with relatively simple dependencies among output variables. As a result, SEAL cannot be directly applied to deterministic 3D HPE settings, which involve far more complex and high-dimensional structural dependencies. Moreover, while SEAL employed generic architectures such as Multi-Layer Perceptrons (MLPs) or BERT (Devlin et al., 2019), 3D HPE calls for a loss architecture tailored to skeletal structure, capable of capturing both local dependencies (adjacent joints, bones) and global dependencies (symmetry, long-range constraints).

### 2.4 GRAPH-BASED MODEL

Graph-based models, such as Graph Convolutional Networks (Kipf & Welling, 2017) and Graph Attention Networks (Veličković et al., 2018), are widely used in human pose estimation because they naturally encode skeletal structure (Zhao et al., 2019; Wang et al., 2024), but their local receptive fields limit long-range reasoning. Graformer addresses this by injecting joint and edge-aware priors into the attention mechanism, enabling global, structure-aware interactions (Zhao et al., 2022). Building on this idea, we design a structure-aware loss-net that guides the pose-net to learn the human kinematic structure consistently.

## 3 METHODOLOGY

### 3.1 PRELIMINARIES: STRUCTURED ENERGY AS LOSS (SEAL)

The Structured Energy As Loss (SEAL) framework was introduced to improve structured prediction by using a structured energy network as a trainable loss function. SEAL trains a secondary network (loss-net) to evaluate the plausibility of predictions. The loss-net provides learning signals to the original pose estimation model (pose-net), enabling it to learn intricate dependencies among outputs without handcrafted rules. In practice, SEAL has shown better performance and fewer constraint violations compared to previous approaches.

Specifically, SEAL has been implemented in two main variants: SEAL-static and SEAL-dynamic. SEAL-static employs a fixed, pre-trained loss-net, whereas SEAL-dynamic continuously updates the loss-net to reflect the evolving outputs of the pose-net. Prior work has shown that SEAL-dynamic generally performs better than SEAL-static by capturing dependencies more effectively and providing stronger guidance during training. Therefore, we integrate the SEAL-dynamic approach into our framework.

### 3.2 SCoTL-POSE

#### 3.2.1 TRAINING PROCEDURE.

Our framework consists of two components: (1) pose-net, which is any 3D HPE model that predicts 3D joint positions from 2D inputs (2) loss-net, a trainable loss function that dynamically learns to evaluate the structural plausibility of predicted poses. The pose-net and loss-net are updated in an alternating manner, allowing the loss-net to dynamically adapt to the evolving predictions of the pose-net and provide structural learning signals.

Specifically, the pose-net is optimized with a combined loss consisting of the standard mean squared error (MSE) and the energy score computed by the loss-net, while the loss-net is jointly trained

---

**Algorithm 1** SCoTL-pose Algorithm

---

**Require:** $(\mathbf{x}, \mathbf{y})$: training data (2D inputs and 3D ground-truth outputs)
**Require:** $F_\phi$: pose-net with parameters $\phi$
**Require:** $E_\theta$: loss-net with parameters $\theta$
**Require:** optimizer$_\phi$: optimizer for pose-net
**Require:** optimizer$_\theta$: optimizer for loss-net
**Require:** $T$: number of training iterations
1: Initialize $\phi_0, \theta_0$ randomly
2: **for** $t = 1$ to $T$ **do**
3:      Sample mini-batch $B_t = \{(x_i, y_i)\}_{i=1}^N$ from training data
4:      Compute pose-net predictions: $\tilde{y}_i = F_{\phi_{t-1}}(x_i)$ for all $x_i \in B_t$
5:      Update loss-net parameters $\theta_t$:
6:         $\theta_t \leftarrow \theta_{t-1} - \eta_\theta \nabla_\theta \frac{1}{|B_t|} \sum_{(x_i,y_i) \in B_t} L_E(x_i, y_i, \tilde{y}_i \, \theta)$
7:      Update pose-net parameters $\phi_t$:
8:         $\phi_t \leftarrow \phi_{t-1} - \eta_\phi \nabla_\phi \frac{1}{|B_t|} \sum_{(x_i,y_i) \in B_t} L_F(x_i, y_i; \theta_t)$
9: **end for**

---

to assign lower energy to ground-truth poses and higher energy to implausible predictions. This iterative optimization enables the pose-net to capture joint dependencies more effectively.

Our framework can be seamlessly combined with various backbone models, from single-frame lifting models to multi-frame temporal models, since the loss-net can be introduced independently of the pose-net architecture. Furthermore, it does not incur any additional inference cost, because the loss-net is only used during training. The overall procedure is summarized in Algorithm 1, and the detailed explanation is shown in Appendix A.1.

### 3.2.2 GRAPH-BASED LOSS-NET.

In SCoTL-pose, the capability of the loss-net to capture structural dependencies in the output space is crucial for guiding the pose-net toward more plausible predictions. Beyond merely modeling local (short-range) and global (long-range) relations, the loss-net must aggregate them into a unified, whole-pose signal that can effectively guide the task network.

To further enhance this ability, we use a graph-based design for the loss-net, enabling more effective use of skeletal structure. We adapt Graphormer (Ying et al., 2021) to our setting—simplified for the task—and use it as the loss-net backbone, retaining its ability to model short- and long-range joint dependencies via self-attention. This design results in a more expressive and structure-aware trainable loss function, providing stronger structural guidance for the task network and ultimately yielding more consistent and coherent 3D pose predictions.

In addition, to assess the suitability of graph-based structures, we also implement an MLP-based loss-net as a baseline. This comparison allows us to examine whether utilizing graph structure provides benefits over a simpler fully-connected neural network. Overall implementation details of loss-nets are provided in Appendix A.2.2.

### 3.2.3 PAIRWISE TEMPORAL LOSS.

While the loss-net can enforce structural plausibility within individual frames, it is required that the loss-net also capture temporal consistency to be effective in multi-frame settings. To this end, we extend plausibility evaluation from the single-frame level to the multi-frame setting, directly improving the temporal consistency and plausibility of predicted poses. Given a sequence of length $N$, we randomly sample start indices $t, s \sim \text{Uniform}\{1, \ldots, N - K + 1\}$ and choose a window length $K$. For each window $W_t = \{t, \ldots, t + K - 1\}$ and $W_s = \{s, \ldots, s + K - 1\}$, we aggregate per-frame energies $E_i$ into a segment energy $\bar{E}(W) = \frac{1}{K} \sum_{i \in W} E_i$. We then incorporate the difference between the two segment energies as a loss term during training, which enhances frame-to-frame consistency.

## 4 EXPERIMENTAL SETUP

### 4.1 DATASETS AND EVALUATION METRICS

**Datasets.** We conduct our empirical experiments on Human3.6M dataset (H36M) (Ionescu et al., 2014), MPI-INF-3DHP (3DHP) (Mehta et al., 2017) dataset and Human3.6M 3D WholeBody dataset (H3WB) (Zhu et al., 2023). H36M is the most widely used dataset for 3D human pose estimation (Zheng et al., 2023; Liu et al., 2024). 3DHP is a more challenging dataset than H36M because it contains fewer samples and includes both indoor and outdoor scenes, while H36M only contains indoor scenes. H3WB is a recent dataset for 3D whole-body pose estimation. H3WB extends H36M by providing whole-body keypoint annotations with detailed information about hands, face, and feet, making it suitable for evaluating fine-grained 3D pose estimation.

**Evaluation Metrics.** We follow common practice in 3D human pose estimation and report standard metrics, such as mean per-joint position error (MPJPE), procrustes-aligned MPJPE (P-MPJPE), percentage of correct keypoints (PCK), area under curve (AUC), and pelvis-aligned MPJPE (PA-MPJPE), according to the evaluation protocol of each dataset. These metrics remain the standard benchmarks to evaluate per-joint error. However, they do not measure structural plausibility, whether the predicted poses conform to anatomical constraints.

### 4.2 STRUCTURAL CONSISTENCY METRICS

To further evaluate structural consistency, we introduce two additional metrics.

**Limb Symmetry Error (LSE).** LSE measures violation of the left–right symmetry by comparing the lengths of the corresponding limbs, such as the lower arms and thighs. For a limb pair $(\mathbf{l}_{i1}, \mathbf{l}_{i2}), (\mathbf{r}_{i1}, \mathbf{r}_{i2})$, LSE is defined as the normalized difference in length between the left and right counterparts:

$$\text{LSE}_i = 100 \cdot \left| \frac{\|\mathbf{l}_{i1} - \mathbf{l}_{i2}\| - \|\mathbf{r}_{i1} - \mathbf{r}_{i2}\|}{(\|\mathbf{l}_{i1} - \mathbf{l}_{i2}\| + \|\mathbf{r}_{i1} - \mathbf{r}_{i2}\|)/2} \right|$$

**Body Segment Length Error (BSLE).** BSLE measures deviations in the lengths of body segments, pair of adjacent joints, by comparing predicted poses and ground-truth poses. A special case of BSLE focuses on limbs, where symmetric differences are most pronounced, is referred to as **limb length error (LLE)**. For each segment $i$, with predicted adjacent keypoints $\mathbf{k}_{i1}, \mathbf{k}_{i2}$ and corresponding ground truth keypoints $\mathbf{t}_{i1}, \mathbf{t}_{i2}$, BSLE is defined as:

$$\text{BSLE}_i = 100 \cdot \left| 1 - \frac{\|\mathbf{k}_{i_2} - \mathbf{k}_{i_1}\|}{\|\mathbf{t}_{i_2} - \mathbf{t}_{i_1}\|} \right|$$

We emphasize that these metrics are not used as training loss. Instead, our trainable loss-net is designed to learn structural consistency directly from data without requiring explicit priors such as fixed bone lengths or symmetry constraints. LSE and BSLE are used solely for evaluation purposes, providing complementary insights into whether predicted poses are anatomically plausible.

### 4.3 BACKBONE MODELS

We evaluate SCoTL-pose under two settings: single-frame and multi-frame 3D human pose estimation. In the single-frame setting, models predict 3D poses from a single frame of 2D keypoints, which is more challenging due to the incomplete visual information. In contrast, the multi-frame setting leverages sequences of frames to exploit richer spatio-temporal cues. We employed widely used pose estimation models as our pose-net backbone. Specifically, we used SimpleBaseline (Martinez et al., 2017), SemGCN (Zhao et al., 2019), and VideoPose (Pavllo et al., 2019) for single-frame setting, and MixSTE (Zhang et al., 2022), P-STMO (Shan et al., 2022), PoseFormerV2 (Zhao et al., 2023), D3DP (Shan et al., 2023) and KTPformer (Peng et al., 2024) for multi-frame setting. These backbones cover a broad range of commonly used architectures, allowing us to verify the effectiveness and robustness of SCoTL-pose across different designs. Further implementation details are provided in the Appendix A.2.

Table 1: **Performances on Human3.6M**. SCoTL-pose improves MPJPE and P-MPJPE across models, using 2D ground-truth keypoints as input in all experiments.

| Method | MPJPE↓ | P-MPJPE↓ |
|---|---|---|
| *Single-frame models* | | |
| SimpleBaseline (Martinez et al., 2017) | 43.8 | 34.7 |
| + SCoTL-pose (MLP) | 42.5 | 33.9 |
| + SCoTL-pose (Graph) | **40.7** | **32.3** |
| SemGCN (Zhao et al., 2019) | 47.0 | 37.9 |
| + SCoTL-pose (MLP) | 44.9 | 36.5 |
| + SCoTL-pose (Graph) | **43.4** | **35.7** |
| VideoPose (Pavllo et al., 2019) | 41.6 | 32.4 |
| + SCoTL-pose (MLP) | **41.0** | 32.3 |
| + SCoTL-pose (Graph) | 41.2 | **32.1** |
| *Multi-frame models* | | |
| MixSTE ($T$=243) (Zhang et al., 2022) | 20.8 | 16.1 |
| + SCoTL-pose (MLP) | 20.6 | 15.8 |
| + SCoTL-pose (Graph) | **20.0** | **15.7** |
| Poseformer V2 ($T$=27) (Zhao et al., 2023) | 42.7 | 31.6 |
| + SCoTL-pose (MLP) | 41.5 | 31.2 |
| + SCoTL-pose (Graph) | **41.2** | **30.7** |
| Poseformer V2 ($T$=27) (Zhao et al., 2023) | 42.7 | 31.6 |
| + SCoTL-pose (MLP) | 41.5 | 31.2 |
| + SCoTL-pose (Graph) | **40.5** | **30.3** |
| D3DP ($T$=243, $H$=20, $K$=10, $J_{Best}$) Shan et al. (2023) | 20.4 | 15.4 |
| + SCoTL-pose (MLP) | 18.1 | 13.9 |
| + SCoTL-pose (Graph) | **17.7** | **13.7** |
| KTPformer ($T$=243, $H$=20, $K$=10, $J_{Best}$) Peng et al. (2024) | 18.9 | 14.3 |
| + SCoTL-pose (MLP) | 18.9 | 14.5 |
| + SCoTL-pose (Graph) | **18.3** | **13.9** |

Table 2: **Performances on MPI-INF-3DHP**. SCoTL-pose consistently reduces MPJPE and improves PCK and AUC, using 2D ground-truth keypoints as input in all experiments.

| Method | MPJPE↓ | PCK↑ | AUC↑ |
|---|---|---|---|
| *Single-frame models* | | | |
| SimpleBaseline (Martinez et al., 2017) | 80.9 | 86.9 | 53.8 |
| + SCoTL-pose (MLP) | 71.8 | 89.3 | 58.7 |
| + SCoTL-pose (Graph) | **68.2** | **90.2** | **60.4** |
| SemGCN (Zhao et al., 2019) | 74.5 | 89.5 | 56.4 |
| + SCoTL-pose (MLP) | 71.8 | 90.4 | 57.9 |
| + SCoTL-pose (Graph) | **62.9** | **92.7** | **61.7** |
| VideoPose (Pavllo et al., 2019) | 66.4 | 90.8 | 60.5 |
| + SCoTL-pose (MLP) | 64.0 | 91.7 | 62.1 |
| + SCoTL-pose (Graph) | **62.2** | **91.8** | **63.1** |
| *Multi-frame models* | | | |
| P-STMO ($T$=81) (Shan et al., 2022) | 34.6 | 97.8 | 76.6 |
| + SCoTL-pose (MLP) | 34.8 | 98.0 | 76.6 |
| + SCoTL-pose (Graph) | **33.8** | **98.1** | **77.3** |
| P-STMO ($T$=81) (Shan et al., 2022) | 33.4 | 98.0 | 77.5 |
| + SCoTL-pose (MLP) | 32.9 | 98.1 | 77.7 |
| + SCoTL-pose (Graph) | **32.5** | **98.2** | **78.0** |
| Poseformer V2 ($T$=27) (Zhao et al., 2023) | 29.6 | 97.0 | 77.8 |
| + SCoTL-pose (MLP) | **28.5** | **97.4** | **78.4** |
| + SCoTL-pose (Graph) | 29.5 | **97.4** | 77.8 |
| Poseformer V2 ($T$=27) (Zhao et al., 2023) | 29.6 | 97.0 | 77.8 |
| + SCoTL-pose (MLP) | 28.5 | 97.4 | 78.4 |
| + SCoTL-pose (Graph) | **27.8** | **97.5** | **79.0** |
| D3DP ($T$=243, $H$=20, $K$=20, $J_{Best}$) Shan et al. (2023) | 28.8 | 98.2 | 80.4 |
| + SCoTL-pose (MLP) | 28.5 | 98.6 | 80.8 |
| + SCoTL-pose (Graph) | **27.8** | **98.7** | **80.9** |

Table 3: **Performance on the Human3.6M WholeBody**. SCoTL-pose reduces P-MPJPE across all body parts, resulting in more coherent predictions. † from H3WB's official benchmark. ‡ nose-aligned MPJPE for face and wrist-aligned MPJPE for hands.

| Method | Whole-body | Body | Face/Aligned‡ | Hand/Aligned‡ |
|---|---|---|---|---|
| Jointformer † | 88.3 | 84.9 | 66.5 / 17.8 | 125.3 / 43.7 |
| 3D-LFM | 64.1 | 60.8 | 56.6 / 10.4 | 78.2 / 28.2 |
| SimpleBaseline | 67.4 | 63.3 | 49.9 / 14.1 | 98.0 / 34.8 |
| + SCoTL-pose (MLP) | **62.8** | **61.1** | **46.3** / 13.7 | **90.7** / 34.2 |
| + SCoTL-pose (Graph) | 64.8 | 61.6 | 47.6 / **13.3** | 94.0 / 34.5 |
| VideoPose | 61.5 | 57.4 | 48.8 / 11.9 | 84.1 / 30.3 |
| + SCoTL-pose (MLP) | **58.6** | **54.8** | **45.0** / 11.5 | **82.3** / 29.3 |
| + SCoTL-pose (Graph) | 59.5 | 56.3 | 46.1 / **11.5** | **82.3** / 28.7 |

# 5 EXPERIMENTAL RESULTS

## 5.1 COMPARISON WITH BASELINES

**Single-Frame Settings.** In the single-frame setting, models predict 3D poses from a single 2D frame with incomplete visual cues, making the task inherently more ambiguous and challenging. As shown in Table 1 (Human3.6M) and Table 2 (MPI-INF-3DHP), SCoTL-pose consistently improves performance across all baseline models. On Human3.6M, SCoTL-pose reduces both MPJPE and P-MPJPE, with graph-based loss-net variants yielding the largest gains. On MPI-INF-3DHP, which contains more diverse and in-the-wild scenarios, the improvements are even more pronounced.

SCoTL-pose also improves whole-body pose estimation on the H3WB dataset, as shown in (Table 3), demonstrating that our trainable loss function is also beneficial in complex whole-body settings. However, when the loss-net is graph-based, its bias toward local neighborhoods may underutilize global context and thus underperform a simpler MLP loss-net. We leave a deeper analysis of the balance between local and global reasoning for future work.

**Multi-Frame Settings.** Multi-frame models already achieve strong performance because they benefit from both advanced backbone designs and the ability to leverage temporal information. For

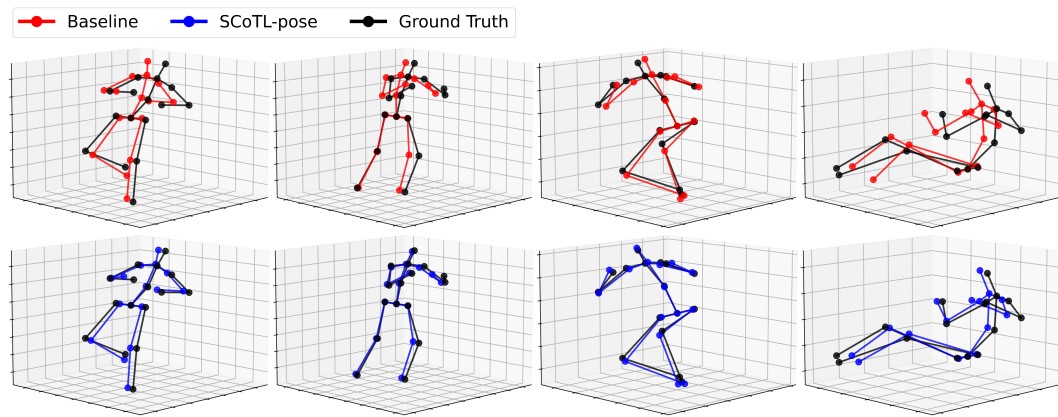

Figure 1: **Qualitative Comparison of Predicted Poses on H36M**. Predictions from SCoTL-pose (bottom, blue) demonstrate clear improvements over the baseline (top, red) by producing structures closer to the ground-truth human pose (black).

Table 4: **Structural Consistency Evaluation Across Datasets**. SCoTL-pose reduces structural error metrics such as LSE, LLE, and BSLE (see §4.2 for the definitions), improving plausibility.

| Dataset | Metric | LSE ↓ | LLE ↓ | BSLE ↓ |
|---------|--------|-------|-------|--------|
| H36M | Ground Truth | 0.00 | 0.00 | 0.00 |
| | SimpleBaseline | 4.85 | 5.09 | 6.12 |
| | + Constraint | 4.35 | 4.54 | 6.17 |
| | + SCoTL-pose (Graph) | **3.68** | **3.94** | **5.49** |
| 3DHP | Ground Truth | 1.21 | 0.00 | 0.00 |
| | SimpleBaseline | 10.14 | 11.60 | 8.13 |
| | + Constraint | 7.80 | 10.02 | 7.87 |
| | + SCoTL-pose (Graph) | **6.22** | **6.02** | **5.93** |
| H3WB | Ground Truth | 4.42 | 0.00 | 0.00 |
| | SimpleBaseline | 6.60 | 6.56 | **6.22** |
| | + Constraint | 6.88 | 6.82 | 6.66 |
| | + SCoTL-pose (Graph) | **6.55** | **6.13** | 6.73 |

instance, architectures such as MixSTE and P-STMO significantly outperform single-frame baselines by exploiting spatial-temporal cues across sequences. Despite their strong baselines, incorporating SCoTL-pose still yields consistent improvements, as shown in Table 1 and Table 2. This indicates that SCoTL-pose complements temporal modeling by providing additional structural guidance, leading to better predictions. Importantly, these gains come without any additional inference cost, highlighting that even state-of-the-art temporal architectures can benefit from a trainable loss function that enforces structural consistency.

## 5.2 STRUCTURAL CONSISTENCY EVALUATION

We evaluated structural consistency by examining the LSE, LLE, and BSLE metrics on the H36M, 3DHP, and H3WB datasets. For comparison, we also included a setting with explicit bone length constraints as a loss term, to directly contrast SCoTL-pose with manually designed constraint-based approaches.

In result, SCoTL-pose consistently showed lower error values across all three structural metrics on H36M and 3DHP datasets, as detailed in Table 4. These results indicate that SCoTL-pose effectively captures structured dependencies in human poses, leading to more anatomically plausible and consistent 3D pose predictions. For a more detailed examination, we grouped samples into bins with comparable P-MPJPE and analyzed our proposed structural consistency metrics (LSE, BSLE, LLE) within each bin. Even when comparing predictions with similar P-MPJPE, SCoTL-

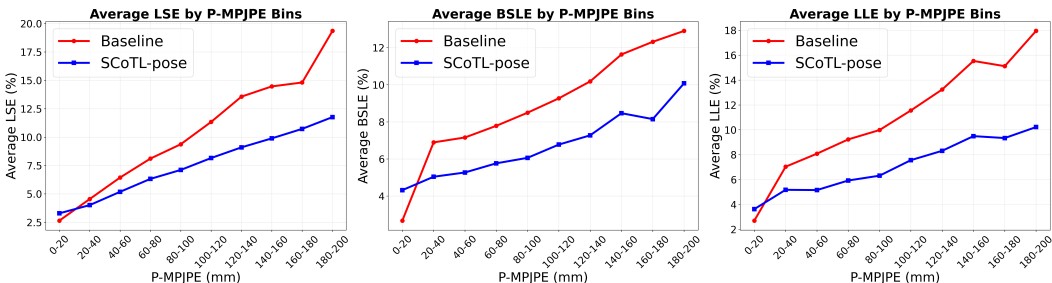

Figure 2: **Comparison of structural consistency in MPI-INF-3DHP.** Average structural inconsistency measures (from left to right: LSE, BSLE, LLE) are displayed for predictions of baselines (red) and SCoTL-pose (blue) binned by P-MPJPE. SCoTL-pose consistently achieves lower structural errors even under similar P-MPJPEs.

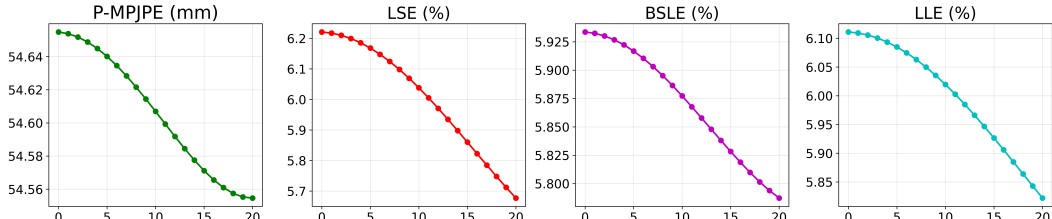

Figure 3: **Gradient-Based Inference results on MPI-INF-3DHP**. P-MPJPE, LSE, LLE, and BSLE all decrease steadily over iterations, indicating that the loss-net effectively captures structural plausibility and provides meaningful corrective feedback to the pose-net.

pose consistently yielded lower values on these metrics than the baseline, as shown in Figure 2. These intra-bin comparisons reveal a limitation of standard training objective: by optimizing primarily for pointwise coordinate error (e.g. MSE, MPJPE), they are relatively insensitive to violations of structural plausibility (e.g., symmetry, limb proportionality, kinematic consistency), leading to higher LSE/BSLE/LLE within similar P-MPJPE ranges. Notably, LSE is defined without reference to ground truth, yet the results on 3DHP highlight that SCoTL-pose more effectively internalizes structural constraints. On Human3.6M, absolute errors are already small, so the differences are less pronounced, but SCoTL-pose still demonstrates an advantage over the baseline.

However, SCoTL-pose showed mixed results on the H3WB dataset, This is likely due to the dataset's noisy labeling, which is shown from the relatively high LSE of the ground truth poses. Moreover, since H3WB provides annotations for a very large number of keypoints including body, face, hands, and feet, capturing coherent structural dependencies across all regions is inherently more challenging, making further improvements less straightforward. While SCoTL-pose did not significantly outperform the baseline in this setting, it still achieved better results compared to directly injecting explicit structural constraints, suggesting that a trainable loss function provides a more flexible and generalizable way of enforcing plausibility in whole-body pose estimation.

Overall, the improved structural consistency metrics highlight that loss-net's ability to capture structures in human poses helps the pose-net to predict more anatomically consistent and plausible 3D human poses.

## 5.3 ABLATION STUDIES

**Gradient-Based Inference with Loss-Net.** We conduct gradient-based inference (GBI) on the output of the pose-net using the trained loss-net to verify its ability to capture plausible human pose structures. GBI iteratively refines the predicted poses by following the gradient signals from the loss-net, which are expected to lower the assigned energy, with details provided in Appendix A.3. As shown in Figure 3, P-MPJPE, as well as the structural metrics LSE, LLE, and BSLE, all steadily decrease over dozens of iterations. Since these metrics directly reflect structural plausibility, the

consistent reduction indicates that the loss-net effectively caputres human pose structure and provides meaningful gradient signals. The effect is more pronounced on the challenging 3DHP dataset but the same trend is also observed on H36M, as shown in Figure 4 in Appendix A.3, confirming the consistency of the results.

**Analysis of Graph-based Architecture.** We compared graph-based and MLP-based loss-nets. Overall experimental results show that the graph-based loss-net consistently provides better structural guidance, leading to lower per-joint errors as well as improved structural plausibility, as shown in Sections 5.1 and 5.2. This advantage stems from the fact that human poses can be naturally represented as graphs, where joints correspond to nodes and bones to edges. By leveraging this inductive bias, the graph-based loss-net can capture both local constraints (e.g., bone lengths, adjacent joint dependencies) and global relationships (e.g., symmetry, cross-limb coordination) that are difficult for MLP-based loss-nets to encode explicitly.

**Effect of Pairwise Temporal Loss.** To investigate the contribution of the proposed pairwise temporal loss, We focus on MixSTE because it is a seq2seq model that predicts the entire input sequence, enabling pairwise comparisons across temporal segments. While the loss-net enforces structural plausibility within each frame, inter-frame consistency is also important for multi-frame settings. The pairwise temporal loss encourages aligning the energy distributions of different temporal segments, thereby reducing abrupt frame-to-frame variations. Emperical results demonstrate that adding the pairwise temporal loss improves per-joint errors and temporal coherence, demonstrating that it promotes temporal consistency across video sequences, as shown in Table 5 in Appendix A.4.

# 6 CONCLUSION

In this paper, we propose SCoTL-pose, a novel framework that introduces a trainable loss function for 3D human pose estimation. Unlike prior approaches relying on explicit priors or architecture-specific constraints, our proposed loss-net learns structural dependencies directly from data, and can be seamlessly integrated with a wide range of backbone pose-nets. We design the loss-net as a graph-based model, representing joints as nodes and bones as edges, which enables principled learning of both local (bone lengths, adjacency) and global (symmetry, long-range relations) dependencies in human pose. Our experiments on Human3.6M, MPI-INF-3DHP, and H3WB demonstrate that SCoTL-pose not only reduces per-joint pose estimation errors, but also improves structural plausibility, confirmed by our proposed LSE and BSLE metrics, showing that the graph-based loss-net provides stronger structural guidance. Overall, SCoTL-pose highlights the promise of trainable loss functions as a general paradigm for structured prediction tasks with complex output dependencies.

# 7 LIMITATIONS

While SCoTL-pose demonstrates clear improvements in 3D human pose estimation, certain limitations remain. A primary challenge lies in the broad hyperparameter search space, which includes the weighting of the energy loss term, learning rates for both the pose-net and loss-net. Such a large search space makes training less straightforward and can increase computational overhead. Developing more systematic strategies for efficient hyperparameter tuning and stable optimization would further enhance the practicality and scalability of SCoTL-pose.

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

# A  APPENDIX

## A.1  DETAILED SCoTL-POSE

In our framework, the pose-net $F_\phi(x)$ is optimized to minimize a weighted sum of the mean squared error (MSE) loss and the output of the loss-net (energy) $E_\theta(x, \tilde{y})$. Specifically, the pose-net parameters $\phi$ are updated using the following manner:

$$\phi_t \leftarrow \phi_{t-1} - \eta_\phi \nabla_\phi \frac{1}{|B_t|} \sum_{(x,y) \in B_t} L_F(\phi; \theta), \tag{1}$$

where $B_t$ is the mini-batch of training samples at iteration $t$, $\eta_\phi$ is the learning rate for the pose-net, and $L_F(\phi; \theta)$ is the combined loss function. The combined loss function is defined as:

$$L_F(x_i, y_i; \theta) = \sum_{j=1}^{M} \text{MSE}(y_j, F_\phi(x)_j) + \alpha E_\theta(x, F_\phi(x)), \tag{2}$$

where $M$ refers to the total number of joints in the pose estimation cdataset and $x$ represents the input data, specifically the 2D joint coordinates. The variable $y_j$ denotes the ground-truth 3D joint coordinates, while $F_\phi(x)_j = \tilde{y}_j$ represents the predicted 3D joint coordinates from the pose-net. The energy term $E_\theta(x, F_\phi(x))$ is computed by the loss-net and implicitly evaluates the structural dependencies between joints. Finally, $\alpha$ is a hyperparameter controlling the balance between the MSE loss and the energy term.

The loss-net is dynamically trained to adapt to the pose-net's predictions by minimizing the loss $L_E$:

$$\theta_t \leftarrow \theta_{t-1} - \eta_\theta \nabla_\theta \frac{1}{|B_t|} \sum_{(x,y) \in B_t} L_E(x, y, F_{\phi_{t-1}}(x); \theta). \tag{3}$$

We employ two types of loss for $L_E$: margin-based loss and a simplified form of noise contrastive estimation (NCE) ranking loss (Ma & Collins, 2018), both suggested in Lee et al. (2022).

The margin-based loss enforces the loss-net to decrease the energy $E_\theta(x, y)$ of the ground truth label $y$ and increase the energy $E_\theta(x, \tilde{y})$ of the pose-net's incorrect prediction $\tilde{y}$, such that the difference between the two energies is sufficiently large to exceed the margin. The margin-based loss is defined as:

$$L_E^{\text{margin}}(x_i, y_i, \tilde{y}_i; \theta) = \max_{\tilde{y}} \left[ \Delta(y, \tilde{y}) - E_\theta(x, \tilde{y}) + E_\theta(x, y) \right]_+, \tag{4}$$

where $\Delta(y, \tilde{y})$ denotes a task-specific margin function, MPJPE in our implementation.

Similarly, the NCE ranking loss minimizes the energy of the ground truth label $y$ while increasing the energy of the pose-net's prediction $\tilde{y}$, treating the pose-net's predictions as negative samples. The NCE ranking loss is defined as:

$$L_E^{\text{NCE}}(x_i, y_i, \tilde{y}_i; \theta) = -\log \frac{\exp(-E_\theta(x, y))}{\exp(-E_\theta(x, y)) + \exp(-E_\theta(x, \tilde{y}))}. \tag{5}$$

## A.2  IMPLEMENTATION DETAILS

### A.2.1  POSE-NET

We have modified the input and output layers of pose-net models to align with the dimensions of each dataset. In single-frame settings, we used separate Adam optimizers (Kingma & Ba, 2015) without learning rate decay for the loss-net and pose-net and trained models with a batch size of 1024 for 50 epochs on H36M and 3DHP, and a batch size of 64 for 200 epochs on H3WB. For multi-frame models, we used reported hyperparameters in their original papers.

### A.2.2  LOSS-NET

Graphormer injects structure into self-attention by adding shortest-path distance (SPD)–based spatial bias $b_{\phi(v_i, v_j)}$ and an edge-path bias $c_{ij}$ (edges along a shortest path) to the attention logits:

$$A_{ij} = \frac{(h_i W_Q)(h_j W_K)^\top}{\sqrt{d}} + b_{\phi(v_i, v_j)} + c_{ij}.$$

Separately, degree-based centrality is encoded by adding learnable embeddings to node inputs (not as an attention bias) (Ying et al., 2021). Because human skeletal graphs are small and regular compared with molecular or social networks, we adopt Graphormer's core idea while simplifying it for human pose estimation. Concretely, we (i) remove node-level degree–centrality embeddings and (ii) do not define categorical edge types. On small, skeletal graphs, such handcrafted encodings can act like noise and distract attention, so we eliminate them and let the model infer structure directly from data while retaining only the shortest-path biases. This minimalist biasing reduces spurious inductive signals and helps the pose-net produce outputs with stronger structural consistency. Beyond encoding local and global dependencies, Graphormer also introduces a global, CLS-like virtual node $v_{\mathrm{cls}}$ that aggregates whole-graph information (Ying et al., 2021). We adopt this component in the loss-net: a virtual node $v_{\mathrm{cls}}$ attends to all joints and produces a compact summary signal of skeletal structure. This design summarizes pose-level structure and guides the pose-net toward structurally coherent outputs.

We design the graph-based loss-net, following the Graphormer foundation, with model width $d{=}256$, $H{=}8$ attention heads, depth = 6 blocks; the same graph bias is shared across layers. For inputs, we represent each node by concatenating the keypoint's 2D coordinates with its predicted 3D coordinates and encoding the joint identity via a one-hot vector; the resulting feature is linearly projected to dimension $d{=}32$, and a learnable CLS token is prepended. The head is an MLP that outputs a scalar energy, and we train the loss-net with either a margin-based objective (e.g., an MPJPE margin) or NCE.

For the MLP loss-net, we adjusted the SimpleBaseline architecture by modifying the dimensions and depth of the hidden layers. Specifically, we set the hidden size to 2048 with 2 residual block stages and omitted batch normalization and dropout layers.

## A.3 Gradient-Based Inference

We implement a gradient-based inference (GBI) method with trained loss-net and pose-net, to examine whether the loss-net effectively captures structural dependencies in human poses. GBI is a method that leverages gradients to iteratively refine the outputs (Goodfellow et al., 2015; Mordvintsev et al., 2015; Gatys et al., 2015b;a; Belanger & McCallum, 2016) or parameters (Lee et al., 2019) of neural networks, and we adopt the former approach. Specifically, we iteratively update the predictions of pose-net along the gradient provided by the loss-net, with the objective of decreasing the energy. This procedure provides a direct way to evaluate whether the learned energy function captures human pose structure. If the loss-net has successfully learned structural dependencies, then following its gradient should progressively refine the predictions toward more plausible poses. The results on the H36M dataset are shown in Figure 4, where all metrics steadily decrease over iterations, consistent with the trends observed for 3DHP in Figure 3.

## A.4 Effect of Pairwise Temporal Loss

Table 5 demonstrates the effect of incorporating pairwise temporal loss. Compared to the baseline MixSTE, SCoTL-pose with pairwise temporal loss achieves better performance, even though the strong baseline leaves limited room for improvement. This indicates that the pairwise temporal loss promotes inter-frame consistency, complementing the structural guidance of the loss-net.

Table 5: SCoTL-pose on H36M with fixed $\Lambda$ and $K$ columns. In parentheses: $\Delta$ vs. MixSTE ($T{=}243$); negative is better.

| Setting | $\Lambda$ | $K$ | $T$ | MPJPE↓ | P-MPJPE↓ | MPJVE↓ |
|---|---|---|---|---|---|---|
| MixSTE (baseline) | — | — | 243 | 20.8 | 16.1 | 0.94 |
| SCoTL-pose (Graph margin) | $10^{-3}$ | — | 243 | 20.3 (-0.5) | 15.8 (-0.3) | 0.96 (+0.02) |
| SCoTL-pose (Graph margin + pair-loss) | $10^{-3}$ | 3 | 243 | **20.0** (-0.8) | **15.7** (-0.4) | **0.93** (-0.01) |

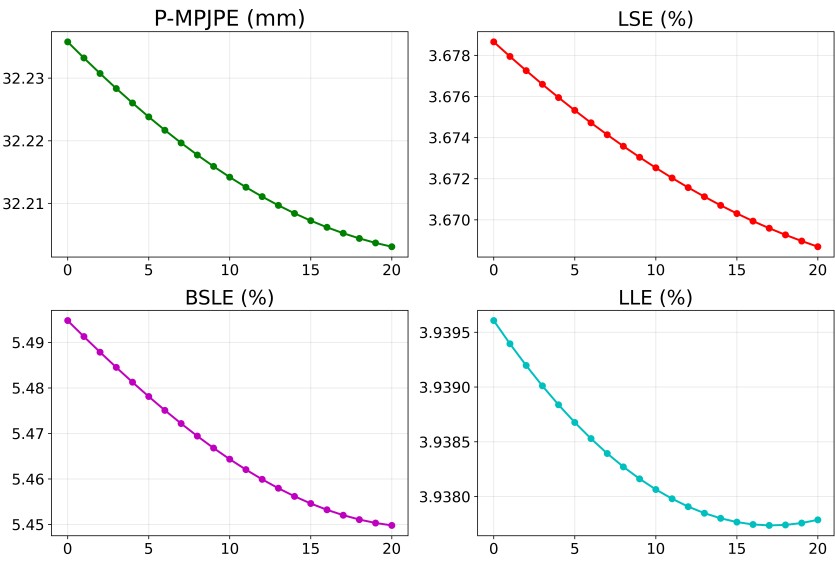

Figure 4: **Gradient-Based Inference results on H36M**. P-MPJPE, LSE, LLE, and BSLE all decrease steadily over iterations, indicating that the loss-net effectively captures structural plausibility and provides meaningful corrective feedback to the pose-net.

