# OpenReview forum: "Enhancing structural consistency of 3D Human Pose Estimation through Trainable Loss Function"
_ICLR.cc/2026/Conference — Submitted to ICLR 2026_

### Official Review · Reviewer_xYLb · 2025-10-31

**Soundness:** 2
**Presentation:** 2
**Contribution:** 1
**Rating:** 4
**Confidence:** 4

**Summary:**

This paper introduces SCoTL-Pose, an extension of the Structured Energy As Loss (SEAL) framework, tailored for the high-dimensional and structurally complex domain of 3D human pose estimation. Additionally, it proposes two metrics—Limb Symmetry Error (LSE) and Body Segment Length Error (BSLE)—and demonstrates performance improvements across the Human3.6M, MPI-INF-3DHP, and H3WB datasets..

**Strengths:**

1. The proposed loss network (loss-net) is employed only during training, thus incurring no additional inference cost.
2. The framework is compatible with various backbone architectures, including SimpleBaseline, VideoPose, and MixSTE. Extensive experiments validate the effectiveness of the proposed method.

**Weaknesses:**

1. This work represents an incremental extension of SEAL-Pose [1], which was accepted at the ICCV 2025 Workshop on SP4V. The primary modification lies in replacing the loss-net with a Graph Network to better capture skeletal topology. While this change offers structural advantages, it is largely engineering-level rather than conceptual or theoretical.
2. The reported performance differences between SCoTL-Pose (MLP) and SEAL-Pose (Margin) are inconsistent—some results improve, while others remain similar. These inconsistencies raise concerns regarding reproducibility and experimental reliability.
3. Evaluate cross-dataset generalization, e.g., training on MPI-INF-3DHP and testing on Human3.6M.
4. Extend the framework to other structured prediction tasks, such as hand pose or animal pose estimation.

[1] SEAL-Pose: Enhancing 3D Human Pose Estimation through Trainable Loss Function.

**Questions:**

See weaknesses

---

> ### Author Response · Authors · 2025-11-23
>
> Dear reviewer xYLb, thank you for your careful reading and insightful comments.
> ## W1: Limited novelty beyond SEAL-pose
>
> We would like to clarify that, beyond replacing the loss-net with a graph network, **the current submission makes several substantial, non-trivial advances:**
>
> **A.** Domain-specific loss-net design. **Our contribution is not merely to “plug-in” of a graph structure.** We carefully designed **(1) how the loss-net inputs (2D/3D poses, joint identity) are preprocessed,** and **(2) the graph-based architecture itself so that it can better exploit skeletal topology in a domain-specific way.** These design choices are discussed in **detail in General Response W1-B.**
>
> **B.** Systematic hyperparameter search protocol. **We conducted a systematic hyperparameter search for the training of pose-net and loss-net.** Detailed settings and concrete examples are provided in **General Response W1-C and General Response W3.**
>
> **C.** **Robustness across diverse backbones.** We instantiated the framework on a broader set of multi-frame backbones than in the workshop version, showing that the learned loss is robust across architectures. We have also begun applying it to more recently found models; **the preliminary results are positive, and we will include them in an updated version.**
>
> **D.** **Analysis of how energy influences the model.** Finally, we added explicit analyses of how the learned energy interacts with model predictions. As shown in Figures 2 and 3 and further discussed in **General Response W4-3,4, we provide empirical and diagnostic analysis of the energy– structural metric relationships and ordering behavior,** which was entirely missing in the earlier workshop paper.
>
> Taken together, **these elements go significantly beyond an engineering-level change** and, we believe, justify viewing this work as a substantial and systematic extension of the learned-loss framework toward human pose estimation rather than a minor modification.
>
> For these reasons, we believe SEAL-Pose [1] is better viewed as an early, exploratory step. At the same time, **the present paper is the first comprehensive and principled treatment of this learned-loss framework for 3D pose estimation,** going substantially beyond a simple application of the loss-net module.
>
> ## W2: Inconsistent performance gaps between SCoTL-Pose (MLP) and SEAL-Pose(Margin)
> We agree that the differences between SCoTL-Pose (MLP) and SEAL-Pose (Margin) can appear inconsistent at first glance. However, these variations are simply the outcome of **the hyperparameter search we designed for this submission.** By training all models and settings under a more systematic searching protocol, we discover better-performing parameters in specific settings while others remain similar. The details of this hyperparameter search are described in **General Response W1-C.**
>
> ## W4: Apply to hand pose or animal pose
> We appreciate this suggestion and agree that extending our framework to other structured prediction tasks (e.g., hand or animal pose estimation) would be very interesting, but leave this research topic for the future work.

---

### Official Review · Reviewer_eSb3 · 2025-10-31

**Soundness:** 2
**Presentation:** 2
**Contribution:** 2
**Rating:** 4
**Confidence:** 4

**Summary:**

This paper proposes a method to enhance 3D Human Pose Estimation (HPE) through structural constraints. The framework consists of two main components:
1）a graph-based trainable loss network
2）two types of evaluation metrics (LSE and BSLE) used to assess the structural quality of predicted poses.
Overall, the proposed structure is applied to some baseline models and leads to certain performance gains. However, the paper’s organization and presentation are not sufficiently clear, and several concerns are noted below:

1. The authors state that the loss network is only required during training and does not introduce additional inference cost. However, it is unclear how the loss network can be used only during training.
2. The Pairwise Temporal Loss is not clearly formulated, and while LSE and BSLE are introduced as evaluation metrics, the training targets used to guide the optimization are not explained.
3. The relationship between the pose network and the loss network is not clearly described. It is recommended that the paper include a diagram illustrating the overall framework, showing how the pose-net and loss-net interact and what specific roles they play in improving the model.
4. The backbone models chosen for comparison are relatively outdated. The proposed SCoTL-Pose framework has not been integrated with more recent strong baselines such as MotionAGFormer or KTPFormer, which limits the significance of the reported performance. Furthermore, the MixSTE results reported in Table 1 appear to be incorrect, as the MPJPE for MixSTE should be around 40.9 mm according to the original paper.
5.The comparative methods listed in the figures and tables should include proper citations to help readers locate and verify the referenced works, as well as the "single frame" or "multi-frame" setting marks.

**Strengths:**

The proposed structure is applied to several baseline models and achieves certain performance gains. It demonstrates a feasible design that can be integrated into both single-frame and multi-frame architectures.

**Weaknesses:**

Refer to the Summary part.

**Questions:**

Refer to the Summary part.

---

> ### Author Response · Authors · 2025-11-20
>
> Dear Reviewer eSb3, We sincerely appreciate your time and effort in reviewing our paper! We provide detailed discussions based on your review
>
> **We revised the wording for clarity; I’d appreciate it if you could take a quick look.**
>
> We will also provide additional responses to the other questions and would greatly appreciate it if you could review them as well.
>
> ## **S1**:Unclear training-only usage of the loss-net
>
> Our loss-net is used purely as a learned loss function, not as part of the standard inference pipeline. **During training**, we first obtain **a prediction from the pose-net** (ŷ = f(x)) and then **evaluate the loss-net energy** E(x, ŷ); this scalar energy is **added to the usual supervised loss** (MSE) and is **only used to backpropagate gradients to update the pose-net and the loss-net parameters.** Once pose-net is trained this way, at test time, **we do not utilize the loss-net at all:** all reported results are obtained by running the pose-net alone, exactly as in the baseline, with no additional forward passes or refinement steps.
>
> The **gradient-based inference (GBI) experiments(Figure3)** are only used as **an analysis tool to visualize the effect of the learned energy model**; they are not part of our deployment-time procedure and are not used to obtain any of the main quantitative results in the paper.
>
> ## **S2-1**:Unclear Pairwise Temporal Loss and missing training targets
>
> Our intention is to compare **short, locally smooth motion segments** and encourage the loss-net to assign **higher energy to segments that are temporally less consistent.**
>
> Concretely, for a sequence of length N, we randomly sample two time indices t and s. Around each index, we take a short temporal window of length k (e.g., frames $[t,t+1,…,t+k−1]$ and $ [s,s+1,…,s+k−1]$). Since the poses inside each window are adjacent in time, the poses within a window are expected to be very similar to each other and represent a single local motion segment.
>
> Formally, let us denote the average pose error (e.g., MPJPE) within each window by
> $D_t$ and $D_s$, and the average loss-net energy within each window by $Ē_t$ and $Ē_s$:
>
> - $D_t = \frac{1}{k} * \sum_{j=0}^{k-1} || ŷ_{t+j} - y_{t+j} ||_2$
> - $D_s = \frac{1}{k} * \sum_{j=0}^{k-1} || ŷ_{s+j} - y_{s+j} ||_2$
> - $Ē_t = \frac{1}{k} * \sum_{j=0}^{k-1} E(x_{t+j}, ŷ_{t+j})$
> - $Ē_s = \frac{1}{k} * \sum_{j=0}^{k-1} E(x_{s+j}, ŷ_{s+j})$
>
> We then define the **margin as the difference in average pose error between the two windows**,
> $m = |D_t - D_s|$,
> and encourage the **difference in average energy to be at least as large as this margin.** A simple margin-based temporal loss can be written as
>
> $L_{temp} = max(0, m - |Ē_t - Ē_s|).$
>
> Intuitively, if two short, locally smooth motion segments differ in their average pose quality (MPJPE), then the gap between their average energies should be “stretched” by at least the same amount. This encourages the **loss-net to reflect temporal pose quality at the level of short segments,** not just individual frames.
>
> We emphasize that this **pairwise temporal loss is not a main contribution** of the paper, but rather an **exploratory attempt to extend SCoTL-pose** toward multi-frame consistency. We treat this pairwise temporal loss as a preliminary design, and we plan to explore more systematic ways of modeling temporal consistency with learned energy as future work.
>
> ## **S4-1**:Outdated baselines
>
> Our intention was to cover a set of **widely used and representative backbones**, from single-frame to multi-frame architectures. Among them, PoseFormerV2 is already a relatively recent transformer-based model and we want the reviewers to note that our approach improves HPE model across dataset, across different models.
>
> Nonetheless, **we fully agree that adding stronger and more recent baselines will further strengthen the empirical evidence** for SCoTL-pose. In parallel to the submission, **we have been integrating SCoTL-pose with a diffusion-based model (D3DP) and with the KTPFormer;** results are positive, and we plan to include these additional experiments in an updated version.

---

### Official Review · Reviewer_TBiB · 2025-11-01

**Soundness:** 2
**Presentation:** 3
**Contribution:** 3
**Rating:** 4
**Confidence:** 4

**Summary:**

The paper proposes SCoTL-pose (Structural Consistency via Trainable Loss for Pose Estimation), a framework to address a key challenge in 3D Human Pose Estimation, i.e., standard losses like MSE optimize for per-joint accuracy but often produce anatomically implausible poses. Instead of using manually designed, rule-based constraints, this work introduces a trainable loss function (loss-net) that learns to assess the structural plausibility of a predicted pose. This loss net is trained jointly with the main pose estimation model in an alternating, dynamic fashion, inspired by the Structured Energy As Loss (SEAL) framework. The authors introduce two evaluation metrics, i.e., Limb Symmetry Error (LSE) and Body Segment Length Error (BSLE) to quantify structural plausibility. Experiments are performed on Human3.6M, MPI-INF 3DHP, and H3WB datasets to show the effectiveness of the proposed approach.

**Strengths:**

1. The design of the loss-net as a graph-based model is a key strength. Using a network architecture that mirrors the output structure, i.e., a skeleton, is an intuitive and good way to enforce structural consistency.
2. The authors validate their method on multiple datasets and six different backbone models (three single-frame, three multi-frame), demonstrating that the model is agnostic of the framework.
3. The introduction of LSE and BSLE as metrics allows for a direct, quantitative evaluation of the plausibility problem, which standard metrics like MPJPE fail to capture. The analysis in Figure 2 shows that SCoTL-pose improves LSE/BSLE even for samples with similar P-MPJPE.
4. Loss-net is only used during training and so adds no computational overhead at test time, which I think is a major practical advantage.

**Weaknesses:**

1. The framework involves an alternating training procedure for two networks, which is essentially a minimax game and similar to GANs which is usually difficult to stabilize. The paper admits this in its limitations ("broad hyperparameter search space," "less straightforward"). I think that this is a major practical weakness.
2. There’s no interpretability or visualization of what the loss-net focuses on. As an example, which limbs or joint dependencies dominate the structural energy? Understanding what the loss is penalizing (symmetry violations, limb length deviations, joint rotations) would make the contribution more interpretable and reliable.
3. The paper mainly compares SCoTL-Pose with vanilla supervised models or simple regularizers. It omits direct comparisons with other structure-aware or constraint-based approaches, such as kinematic tree priors, limb or bone length constraints, graphical model based pose estimators (example Pose-GCNN, kinematic refinement modules). Without such comparisons, it’s hard to understand whether the learned loss is actually superior to explicit structure enforcement.
4. Since the learned loss may rely on dataset specific geometry such as Human3.6, this limits claims of broad generalizability.
5. The temporal pairwise loss is an appealing idea but the results in Table 5 in the appendix aren’t convincing. The improvement is marginal.

**Questions:**

1. Please clarify the "+ Constraint" baseline in Table 4.
2. The paper states SEAL cannot be directly applied. However, the described method (alternating training, margin/NCE loss) appears to be a direct implementation of the SEAL dynamic framework. Could you clarify if the primary novelty is the application of SEAL to this complex regression task and the novel graph-based architecture for the loss-net, rather than a simple modification of the SEAL framework itself?
3. Please answer points arising from the Weakness section as well.

---

> ### Author Response · Authors · 2025-11-23
>
> Dear reviewer TBiB, thank you for your careful reading and insightful comments.
>
> ## W1: Training instability is the major practical weakness of SCoTL-pose.
>
> We acknowledge that alternating optimization can be difficult to stabilize, especially in fully adversarial settings such as GANs. **However, SCoTL-pose showed stable training, almost always outperforming baselines, without model collapsing or catastrophic divergence.**
>
> As we detail in **General Response W1.C, we use a simple two-stage hyperparameter search protocol** that fixes the pose-net learning rate and searches only over the loss-net learning rate and the energy weight. **General Response W3** further provides concrete examples of this search. As shown there, **almost all reasonable combinations improve over the baseline.** Empirically, we therefore **have not found alternating training to be a major practical weakness.**
>
> Moreover, unlike a fully adversarial discriminator in GANs, our loss-net sees a broader input space of pose pairs; please refer to **General Response W3** for a more detailed discussion.
>
> In other words, **the loss-net is encouraged to learn a global structural energy surface** that reflects symmetry and bone-length consistency, rather than overfitting to individual examples. Consequently, **the pose-net is not pushed to move adversarially in the opposite direction of the loss-net,** and the **dynamics are much milder than in GAN-style training.**
>
> ## W2: No interpretability or visualization of what the loss-net focuses on.
>
> We conduct four complementary analyses to demonstrate that the loss-net indeed benefits training:
>
>  (1) Examination of the gradient of the loss-net energy,
>
>  (2) Comparison of predictions with matched P-MPJPE,
>
>  (3) Correlation between energy values and structural inconsistency, and
>
>  (4) Further insight into the differences between the MLP and graph-based energy networks.
>
> **Analyses (1) and (2) were already included in the original submission, while (3) and (4) were newly added during the rebuttal period.** Taken together, **these results show that the loss-net provides meaningful training signals that steer the pose-net toward higher structural consistency,** and they also explain why the graph-based architecture is more effective than an MLP.
>
> Please refer to **General Response W4** for further details.
>
> ## W3: Comparing with the explicit constraint-based method(+Q1)
>
> **First, please refer to General Response 3. If anything is still unclear, we provide a more detailed explanation below.**
>
> In fact, we do include such a baseline in Table 4: the **“+Constraint” variant corresponds to adding bone length & symmetry constraint on top of the vanilla supervised objective, following Xin Cao et al. (2020).**
> In other words, we keep the backbone unchanged and add lightweight penalties that encourage plausible bone lengths and joint configurations.
> $ L_{\text{bone}} = \sum_{k=1}^{K-1}|| \\hat{B}_{k}-B_k^{\\text{gt}} ||_2 + \sum_1 ^ {4}|| \\hat{B}_L - \\hat{B}_R ||_2 $
>
> **We will clarify this implementation more explicitly in the revised version.** Our intention in adding the **“+Constraint” baseline is not to argue that SCoTL-Pose is strictly superior to explicit constraint-based methods.** Instead, we use it to check what level of structural improvement can be achieved when reasonably designed constraints are added on top of the same backbone. In this sense, “+Constraint” provides a reference upper line for the performance that explicit constraints can offer in our setting.
>
> Within this comparison, **Table 4 shows that SCoTL-Pose attains structural metrics that are broadly comparable to, and in some cases slightly better than, this “+Constraint” baseline.**
> We do not take this as evidence that explicit constraints are unnecessary, but rather as an indication that a data-driven loss-net can capture skeletal structure at a level that is competitive with manually specified penalties.
> ## W5: Effect of pairwise loss is marginal
>
> **Even without this pairwise temporal loss, SCoTL-Pose already performs well;** the temporal term was added only to probe whether we could further improve multi-frame consistency. We emphasize that this pairwise temporal loss is not a core contribution but a preliminary, **exploratory extension toward temporal consistency,** and we plan to investigate more systematic temporal consistency modeling in future work.
>
> ## Q2: Limited novelty beyond SEAL
>
> Our contribution is to turn SEAL’s “structure-as-loss” idea into 3D HPE–specific loss-net—with concrete input/architecture design and a simple tuning protocol—that consistently improves diverse backbones without test-time overhead.
> **Please refer to General Response W1 (detailed version): “Novelty beyond SEAL.**

---

### Official Review · Reviewer_7hLc · 2025-11-10

**Soundness:** 3
**Presentation:** 3
**Contribution:** 2
**Rating:** 4
**Confidence:** 4

**Summary:**

This paper proposes SCoTL, the framework introducing a trainable loss network, loss-net, for 3D human pose estimation. Unlike prior approaches relying on manual priors or rule-based constraints, SCoTL-pose learns structural dependencies from data. Extensive experiments on Human3.6M, MPI-INF-3DHP, and Human3.6M WholeBody shohw consistent improvements in both single- and multi-frame settings.

**Strengths:**

1. Comprehensive experiments across multiple datasets and backbones.
2. Well-written and easy to understand; clear motivation and methodology.
3. No additional inference cost, making the approach practical for integration.

**Weaknesses:**

1. Limited novelty beyond SEAL. The main contribution lies in applying a known concept (trainable energy-based loss) to 3D pose estimation, with relatively straightforward modifications.
2. Lack of theoretical analysis: The paper would benefit from a more rigorous examination of why the learned energy function improves plausibility or generalizes.
3. Training instability and sensitivity: The paper acknowledges a large hyperparameter search space but provides little guidance or empirical analysis of its effect.

**Questions:**

1. How sensitive are the results to the α coefficient balancing MSE and the learned energy term?
2. Could the loss-net trained on one dataset transfer to another without re-training?
3. How does this approach compare with explicit constraint-based or manifold-regularized methods in terms of efficiency and robustness?
4. SImilar to the first question, could the loss-net overfit to dataset-specific skeletal proportions or noise patterns?

---

> ### Author Response · Authors · 2025-11-20
>
> Dear Reviewer 7hLc, thank you very much for your careful review of our paper and thoughtful comments! We hope the following responses can help clarify potential misunderstandings and alleviate your concerns.
>
> **We revised the wording for clarity; I’d appreciate it if you could take a quick look.**
>
> ## **W1**:Limited novelty beyond SEAL
> - Bridge from generic SEAL to real-world structured prediction
> - Simple two-stage hyperparameter tuning (fixed pose-net $lr_p$; search over loss-net $lr_l$ and energy weight $\alpha$)
> - Robustness: validated across multiple backbones & dataset
>
> Please refer to the detailed version **General response W1:Regarding novelty beyond SEAL**
>
> ## **W2**:Lack of analysis of the energy model
>
> To demonstrate that the learned loss-net provides **meaningful guidance for the pose**, we introduce a **gradient-based inference (GBI)** as an analysis tool (not as part of our standard inference pipeline). Starting from the pose predicted by the pose-net, we take a few steps of gradient descent on the loss-net energy.
>
> **Figure 3 shows that as the number of GBI iterations increases, P-MPJPE as well as all our structural metrics consistently decrease.**
>
> In other words, when we **follow the gradient of the learned energy function**, the pose moves toward configurations that are both **closer to the ground truth and structurally more consistent.**
>
> To examine this relationship in more detail, we further controlled for P-MPJPE. We stratified samples into bins of similar P-MPJPE, and then **analyzed the association between the energy and structural metrics within each bin**. For **the graph-based loss-net**, the **Kendall τ between energy and LSE is around 0.24–0.26** across all P-MPJPE bins. This corresponds to roughly **62–63%** probability that, within a pair of poses with similar P-MPJPE, **the pose with worse structure receives higher energy**. For BSLE and LLE, this probability similarly rises above 60% in the higher-error bins.
>
> For **the MLP-based loss-net**, the Kendall τ values are noticeably **smaller** in the low- and mid-error bins and only gradually increase in the highest-error bin, indicating a **less consistent alignment between energy and structural metrics.**
>
> Taken together, these results show that **the graph-based loss-net tends to learn an energy that is more consistently ordered with respect to structural errors**, while the **MLP-based loss-net exhibits a comparatively weaker coupling between energy and structure.**
>
> Please refer to the detailed version **General response W4:Lack of analysis of the energy model**
>
> ## **W3**: Training instability & hyperparameter search (+Question1)
>
> When we wrote that our method has a “broad hyperparameter search space”, we did not mean that the optimization is numerically unstable or overly fragile. **Rather, this simply reflects that there are several hyperparameters that we need to jointly tune** — namely, the pose-net learning rate ($lr_p$), the loss-net learning rate ($lr_l$), and the energy weight ($\alpha$).
>
>
> In practice, we adopt a **simple greedy tuning protocol:**
>
> (i) We first tune the pose-net learning rate $lr_p$ (pose-net w/o loss-net),
>
> (ii) Given fixed $lr_p$, we explore the loss-net learning rate $lr_l$ and energy weight $\alpha$ then fix the best $lr_l$.
>
> (iii) Given fixed $lr_p,\ lr_l$, we finally tune the energy weight $\alpha$.
>
> We share that when we apply the above greedy search, **we did not observe “any” training instability, and we saw that SCoTL-pose is not particularly sensitive to energy weight $\alpha$.**
>
> The following With a concrete example on 3DHP with VideoPose shows the specific instance of greedy search over three hyperparameters ($lr_p,\ lr_l,\ \alpha$).
>
> Please refer to **Table. 3DHP with VideoPose we performed a greedy search over three hyperparameters**
>
> The best configuration we found was:
>
> - pose-net lr ($lr_p$) = 1e-4
> - loss-net lr ($lr_l$) = 1e-4
> - energy weight ($\alpha$) = 5e-3
>
>
> This setting yields a MPJPE of 62.18 (< 66.4 of baseline). **Many nearby settings achieve very similar performance, within about 0.7 mm of the best score.** Across the broader grid of learning rate and energy weight, **the resulting MPJPE varies smoothly from roughly 62 mm up to approximately 67 mm as we move away from this well-tuned region,** rather than collapsing or diverging.
>
> Regarding the specific question on the sensitivity to energy weight: for fixed learning rates $lr_p$ = $lr_l$ = 1e-4, **sweeping $\alpha$ over two orders of magnitude (from 1e-4 to 1e-2) changes MPJPE from 62.18 mm to at most 63.82 mm.** We find that **energy weight $\alpha$** in the range of roughly **2e-3–5e-3 gives near-optimal performance**, and values outside this band shift model smoothly rather than catastrophically. Similar patterns hold for other backbones and datasets.

---

> ### Author Response · Authors · 2025-11-20
>
> ## **Q3**:Comparing with the explicit constraint-based method
>
> We want to let the reviewer know that **this comparison is already present: the “+Constraint” variant in Table 4 corresponds to adding bone-length and symmetry constraints** on top of the vanilla supervised objective, following Xin Cao et al. (2020). We will clarify this more explicitly in the revision.
>
> **Our goal is not to claim that SCoTL-Pose is universally superior to explicit constraint-based or manifold-regularized methods.** Instead, we treat them as a strong reference point (an approximate upper bound), and, surprisingly, we find that SCoTL-pose outperforms previous work that added explicit constraints.
>
> Please refer to the detailed version **General response W2**
>
> **Table. 3DHP with VideoPose we performed a greedy search over three hyperparameters**
> | pose-net lr($lr_p$) | loss-net lr($lr_l$) | energy weight($\alpha$) | MPJPE |
> |---|---|---|---|
> | 0.00010 | 0.00010 | 0.005 | 62.178 |
> | 0.00010 | 0.00010 | 0.002 | 62.308 |
> | 0.00005 | 0.00003 | 0.004 | 62.476 |
> | 0.00005 | 0.00003 | 0.005 | 62.505 |
> | 0.00010 | 0.00003 | 0.004 | 62.602 |
> | 0.00010 | 0.00003 | 0.005 | 62.719 |
> | 0.00010 | 0.00010 | 0.001 | 62.820 |
> | 0.00010 | 0.00005 | 0.001 | 62.841 |
> | 0.00010 | 0.00004 | 0.003 | 62.892 |
> | 0.00010 | 0.00005 | 0.005 | 62.919 |
> | 0.00010 | 0.00010 | 0.010 | 62.948 |
> | 0.00010 | 0.00004 | 0.005 | 62.955 |
> | 0.00010 | 0.00004 | 0.004 | 62.998 |
> | 0.00010 | 0.00010 | 0.00050 | 63.072 |
> | 0.00010 | 0.00005 | 0.002 | 63.079 |
> | 0.00010 | 0.00010 | 0.003 | 63.085 |
> | 0.00010 | 0.00002 | 0.005 | 63.236 |
> | 0.00010 | 0.00002 | 0.001 | 63.276 |
> | 0.00005 | 0.00010 | 0.005 | 63.356 |
> | 0.00005 | 0.00010 | 0.004 | 63.365 |
> | 0.00010 | 0.00003 | 0.001 | 63.381 |
> | 0.00010 | 0.00003 | 0.003 | 63.479 |
> | 0.00010 | 0.00001 | 0.005 | 63.520 |
> | 0.00010 | 0.00002 | 0.004 | 63.554 |
> | 0.00005 | 0.00050 | 0.004 | 63.561 |
> | 0.00010 | 0.00010 | 0.004 | 63.570 |
> | 0.00010 | 0.00003 | 0.002 | 63.580 |
> | 0.00020 | 0.00003 | 0.004 | 63.693 |
> | 0.00010 | 0.00005 | 0.004 | 63.782 |
> | 0.00050 | 0.00010 | 0.001 | 63.789 |
> | 0.00010 | 0.00010 | 0.00010 | 63.819 |
> | 0.00020 | 0.00005 | 0.001 | 63.895 |
> | 0.00005 | 0.00003 | 0.00050 | 64.006 |
> | 0.00005 | 0.00005 | 0.001 | 64.045 |
> | 0.00010 | 0.00004 | 0.006 | 64.092 |
> | 0.00010 | 0.00001 | 0.001 | 64.159 |
> | 0.00005 | 0.00003 | 0.00010 | 64.495 |
> | 0.00050 | 0.00003 | 0.004 | 64.555 |
> | 0.00005 | 0.00020 | 0.004 | 64.599 |
> | 0.00005 | 0.00003 | 0.001 | 64.639 |
> | 0.00005 | 0.00003 | 0.00005 | 64.751 |
> | 0.00005 | 0.00003 | 0.00020 | 64.948 |
> | 0.00005 | 0.00003 | 0.002 | 65.091 |
> | 0.00020 | 0.00003 | 0.005 | 65.358 |
> | 0.00050 | 0.00010 | 0.004 | 65.571 |
> | 0.00050 | 0.00003 | 0.005 | 66.290 |
> | 0.00050 | 0.00010 | 0.005 | 67.015 |
> | 0.00002 | 0.00003 | 0.004 | 67.233 |

---

### Author Response · Authors · 2025-11-20
**1st General response**

## W1:Regarding novelty beyond SEAL(7hLc, TBiB, xYLb)

We sincerely thank all reviewers for their thoughtful and detailed feedback, especially for the recurring questions regarding the novelty and positioning of our method relative to SEAL. Below, we clarify how our work concretely extends and instantiates the SEAL paradigm in the context of 3D human pose estimation.

We fully agree that our work is conceptually inspired by the “structure-as-loss” idea introduced by SEAL. **SEAL, however, is a generic framework:** it shows that a trainable energy-based loss can, in principle, enforce structural plausibility, but it **does not provide a concrete recipe for complex, continuous, high-dimensional regression tasks** such as 3D human pose estimation. **Our contribution is to instantiate and validate this paradigm in a genuinely challenging, real-world setting.**

Concretely,

**A.** We are the first to apply a SEAL-style trainable loss to 3D human pose estimation and show that it works **robustly across multiple datasets and six backbones without any test-time overhead.** We are also adding results on two additional strong backbones — a diffusion-based model and the state-of-the-art KTPFormer — which demonstrate that the proposed loss-net is also compatible with high-performing HPE architectures.

**B.** To design the loss-net in a 3D HPE–specific way, we explicitly address two questions: **(1) how to represent and feed the inputs to the loss-net,** and **(2) what architecture to use for mapping pose to a scalar energy.** We first study how to feed both 2D keypoints and lifted 3D keypoints into the energy model, exploring different ways of combining them. In our experiments, **directly concatenating the 2D and 3D keypoints as joint-wise features turns out to be the most effective choice.** We also compare several architectures for the loss-net (MLP, Transformer, and graph-based networks such as GCN/GAT), and **ultimately adopt a graph-based design that explicitly follows the human skeleton topology** while aggregating global pose information into a single scalar energy. **None of these design choices are specified in SEAL;** instead, we introduce substantial **domain-specific adaptations that make SEAL practical for 3D human pose estimation.**

**C.** We also propose a simple but **systematically effective hyperparameter tuning protocol in training**: (1) We keep **the pose-net learning rate fixed to be identical to the baseline (pose-net w/o loss-net) setting.** (2) With the pose-net learning rate fixed, we then **explore several combinations of the loss-net learning rate and the energy weight.** From this stage, we select the **loss-net learning rate that yields the best validation performance,** and finally perform a **focused search over the energy weight alone.** While this greedy hyperparameter search **does not guarantee a globally optimal configuration,** in practice it tends to yield a reasonably reliable operating point **in our experiments, and we haven’t seen more erratic/unstable behaviors compared to the baseline models without loss-net.**

We will revise the main contributions to make this positioning clearer: our goal is not to propose an entirely new generic framework on top of SEAL, but to **bridge the SEAL to a challenging real-world structured prediction task.**

## W2: Comparing with the explicit constraint-based method(7hLc, TBiB, eSb3 )

We would like to clarify the purpose of the +constraint experiments in Table 4. **Our structural metrics are specifically designed to evaluate pose symmetry and bone-length consistency.** Accordingly, we implemented explicit constraints that directly **penalize bone symmetry violations and bone-length deviations, following the formulation of Xin Cao et al. (2020).**

$ L_{\text{bone}} = \sum_{k=1}^{K-1}|| \\hat{B}_{k}-B_k^{\\text{gt}} ||_2 + \sum_1 ^ {4}|| \\hat{B}_L - \\hat{B}_R ||_2 $



As we also explained in our responses to the individual reviewers, **the goal of these experiments was not to argue that our method is inherently superior to adding explicit constraints,** but rather to approximate an upper bound for our method, that does not inject explicit knowledge on how much improvement one might obtain by directly optimizing these structural properties. Somewhat surprisingly, however, we found that our pose estimator **learned with loss-net frequently outperforming application of direct constraints on most structural metrics.**

---

> ### Author Response · Authors · 2025-11-23
> **2nd General response**
>
> ## W3: Training instability & hyperparameter search(7hLC, TBiB)
> We acknowledge that our alternating optimization can seem difficult to stabilize, especially when fully adversarial settings such as GANs are known to be difficult. **However, we would like to notify the reviewers that, in practice, a greedy hyperparameter search (see general response W1.C) on SCoTL-pose showed stable training, almost always outperforming baselines, without model collapsing or catastrophic divergence.**
>
> There could be several reasons behind this. 1) the energy function $E(x,y)$ is defined **over the entire input–pose space,** so in principle **it can compare arbitrary pairs $(x_i,y_i)$ and $(x_j,y_j)$ and induce a global ordering over 3D configurations,** rather than overfitting to a small set of case-specific comparisons.2) we do not have fully adversarial loss as we employ structural loss such as margin-based loss. Nonetheless, we share the following example to display the stability of SCoTL-pose’s training procedure.
>
> The following with a concrete example on 3DHP with VideoPose shows the specific instance of greedy search over three hyperparameters ($lr_p,\ lr_l,\ \alpha$).
>
> The best configuration we found was:
>
> - pose-net lr ($lr_p$) = 1e-4
> - loss-net lr ($lr_l$) = 1e-4
> - energy weight ($\alpha$) = 5e-3
>
> This setting yields a MPJPE of 62.18 (< 66.4 of baseline). Many nearby settings achieve very similar performance, within about 0.7 mm of the best score. Across the broader grid of learning rate and energy weight, **the resulting MPJPE varies smoothly from roughly 62 mm up to approximately 67 mm as we move away from this well-tuned region, rather than collapsing or diverging.**
>
> **Regarding the specific question on the sensitivity to energy weight:** for fixed learning rates, pose-net lr = loss-net lr = 1e-4, sweeping alpha over two orders of magnitude (from 1e-4 to 1e-2) changes MPJPE from 62.18 mm to at most 63.82 mm. We find that energy weight in the range of roughly 2e-3–5e-3 gives near-optimal performance, and values outside this band shift model smoothly rather than catastrophically. Similar patterns hold for other backbones and datasets.

---

> ### Author Response · Authors · 2025-11-23
> **3rd General response**
>
> ## W4:Lack of analysis of the energy model(7hLc, TBiB)
> Analyses (1)–(2) were already included in the original submission, while (3)–(4) were newly added during the rebuttal period.
>
> **[1.Examination of gradient of loss-net energy.]**
>
> To demonstrate that the learned loss-net provides **meaningful guidance for the pose,** in Figure 3, we introduced **gradient-based inference (GBI) as an analysis tool** (not as part of our standard inference pipeline). Starting from the pose predicted by the pose-net, we take a few steps of gradient descent on the loss-net energy.
>
> **Figure 3 (GBI analysis) shows that as the number of GBI iterations increases, P-MPJPE as well as all our structural metrics (LSE, BSLE, LLE) consistently decrease.**
> In other words, when **we follow the gradient of the learned energy function,** the pose moves toward configurations that are both **closer to the ground truth and structurally more consistent** (fewer symmetry violations, more stable limb lengths). This already indicates that the loss-net is not simply acting as a generic regularizer, but is aligned with our explicit notions of structural plausibility.
>
>
> **[2. Comparison of predictions with matched P-MPJPE]**
>
> To ensure that **this effect is not only driven by improving changes in MPJPE,** we also compare predictions of SCoTL-Pose to those of the baseline under matched P-MPJPE. In Figure 2, we focus on samples where the baseline and SCoTL-Pose have similar P-MPJPE and then examine their structural inconsistency metrics.metrics. The SCoTL-Pose predictions tend to have lower LSE/BSLE/LLE than the baseline, albeit having similar P-MPJPE values..
> This suggests that the **loss-net has learned to distinguish structurally better and worse poses beyond what can be explained by MPJPE alone,** and that its energy meaningfully refines the ordering of poses within the same error level(P-MPJPE).
>
>
> **[3. Correlation between energy value v.s. Structural inconsistency]**
>
> We then examine this relationship **in a more fine-grained way by explicitly conditioning on P-MPJPE.** Specifically, we stratify samples into bins of similar P-MPJPE and, within each bin, analyze **the correlation association between the energy and each structural metric.**
> For the graph-based loss-net, **the Kendall τ between energy and LSE is around 0.24–0.26 across all P-MPJPE bins.**
> This corresponds to roughly a **62–63% probability** that, within a pair of poses with similar P-MPJPE, the pose with worse symmetry receives higher energy.
> We observe positive correlations for BSLE and LLE as well, and in the higher-error bins, the corresponding probabilities similarly rise above 60%. Importantly, however, the correlation with LSE is consistently the strongest among the three metrics, **indicating that the learned energy is most sensitive to bone symmetry violations** (e.g., mismatched left–right limbs). **This directly addresses which joint dependencies dominate the energy: the loss-net particularly emphasizes symmetric limb pairs and their relative distances. (per reviewer 7hLc, TBiB’s question)**
>
> **[4. Further insight between MLP and Graph-based energy network]**
>
> For the MLP-based loss-net, **the Kendall τ values are noticeably smaller in the low**- and mid-error bins and only gradually increase in the highest-error bin, indicating a less consistent alignment between energy and structural metrics. When we directly compare the graph-based and MLP-based loss-net, we find that the graph-based loss-net achieves substantially **higher correlation with all three structural metrics** (on the order of ~0.1 absolute Kendall τ across bins). In other words, **encoding the skeletal graph makes the loss-net’s energy much more tightly ordered with respect to symmetry** and limb-length than an MLP with no explicit relational structure.
>
> **[Summary]**
>
> Taken together, these analyses—GBI behavior (Figure 3), matched-MPJPE comparisons between SCoTL-Pose and the baseline (Figure 2), and fine-grained correlation studies across structural metrics and architectures—**provide an interpretable picture of what the loss-net focuses on.** The learned energy is not a black-box scalar: **it systematically tracks structural inconsistencies, with a particularly strong emphasis on symmetry,** and does so more reliably in the graph-based loss-net than in the MLP-based loss-net.

---

### Author Response · Authors · 2025-12-03
**Supplementary Website with Comprehensive Responses and Additional Analyses**

We thank the reviewers for their constructive feedback, which prompted us to perform additional analyses and experiments that more clearly highlight the strengths of SCoTL-Pose.

This website (https://somerandomuser1125.github.io/scotlpose_rebuttal.github.io/) **(1)** outlines the overall architecture of SCoTL-Pose, **(2)** demonstrates consistent gains on both general and SOTA backbones, **(3)** visualizes predictions on unseen in-the-wild data to highlight structural plausibility, **(4)** shows that the loss-net provides effective guidance during training, and **(5)** reports cross-dataset experiments that directly probe potential overfitting of the loss-net.

For more details, please refer to our anonymous supplementary website linked above.

---

### Meta-Review · Area_Chair_A4GG · 2026-01-09

**Summary:**

This paper proposes a method for enhancing 3D HPE through structural constraints. The framework consists of two main components: a graph-based trainable loss network, and two types of evaluation metrics (LSE and BSLE) used to assess the structural quality of predicted poses.  Extensive experiments across multiple datasets and backbones are appreciated.  However, the reviewers raised concerns regarding limited novelty against SEAL and SEAL-pose, lack of detailed analysis of the proposed method, unclear training instability and sensitivity, and generalizability.  In the rebuttal, the authors tried to address the concerns.  AC thinks that some were resolved while some remains.

The provided analysis of the energy model is detail and almost convincing while decrease of P-MPLPE and structure metrics along GBI iterations seems marginal, meaning pose movement toward configurations closer to GT and structurally more consistent may be limited.  Although a greedy hyperparameter search shows stable training, this approach is empirical and ad hoc.  Some reasons to support this approach are argued, but more rigorous justification to guarantee stability is required. Novelty beyond SEAL is intensively argued; however, novelty beyond SEAL-pose is limited.  AC acknowledges non-trivial advance from SEAL-pose indeed.  But the advance is limited but not ground-breaking.  As described above, the proposed systematic hyperparameter search protocol is not well justified, giving the impression of engineering innovation. Robustness analysis across diverse backbones is in progress.  No arguments are provided regarding generalizability.  Taking these in to account, AC thinks that remaining concerns outweigh the technical innovation, leading to limited contribution of this paper.  The paper will be beneficial from substantial improvement. For this, this paper cannot be accepted.

**Reviewer Concerns:**

The main concerns are on limited novelty, lack of analysis of the energy model, missing arguments on training stability, and generalization capability.  The analysis of the energy model was detailed in a convincing way. The other concerns were not adequately addressed as explained above.

**Reviewer Scores:**

Three reviewers 7hLC, TBiB, and xYLb would keep their initial scores because of limited novelty, less rigorous arguments on training instability, and unaddressed generalizability.  Reviewer eSb3 might increase the score to 6 because most of his/her concerns were resolved.

---

### Decision · Program_Chairs · 2026-01-26

Reject